# Enhanced Bioactive Potential of Functionalized Injectable Platelet-Rich Plasma

**DOI:** 10.3390/molecules28041943

**Published:** 2023-02-17

**Authors:** Emoke Pall, Alexandra Roman, Diana Olah, Florin Ioan Beteg, Mihai Cenariu, Marina Spînu

**Affiliations:** 1Department of Clinical Sciences, University of Agricultural Sciences and Veterinary Medicine, 400374 Cluj-Napoca, Romania; 2Department of Periodontology, Faculty of Dental Medicine, Iuliu Haţieganu University of Medicine and Pharmacy, 400347 Cluj-Napoca, Romania

**Keywords:** injectable platelet-rich plasma, human lactoferrin, proliferation, antimicrobial, antibiofilm

## Abstract

Injectable platelet-rich fibrin (iPRF) is a frequently used platelet concentrate used for various medical purposes both in veterinary and human medicine due to the regenerative potential of hard and soft tissues, and also because of its antimicrobial effectiveness. This in vitro study was carried out to assess the cumulative antimicrobial and antibiofilm effect of iPRF functionalized with a multifunctional glycoprotein, human lactoferrin (Lf). Thus, the ability to potentiate cell proliferation was tested on keratinocytes and evaluated by the CCK8 test. The combinations of iPRF and Lf induced an increase in the proliferation rate after 24 h. The average cell viability of treated cultures (all nine variants) was 102.87% ± 1.00, and the growth tendency was maintained even at 48 h. The highest proliferation rate was observed in cultures treated with 7% iPRF in combination with 50 µg/mL of Lf, with an average viability of 102.40% ± 0.80. The antibacterial and antibiofilm activity of iPRF, of human lactoferrin and their combination were tested by agar-well diffusion (Kirby–Bauer assay), broth microdilution, and crystal violet assay against five reference bacterial strains. iPRF showed antimicrobial and antibiofilm potential, but with variations depending on the tested bacterial strain. The global analysis of the results indicates an increased antimicrobial potential at the highest concentration of Lf mixed with iPRF. The study findings confirmed the hypothesized enhanced bioactive properties of functionalized iPRF against both Gram-positive and Gram-negative biofilm-producing bacteria. These findings could be further applied, but additional studies are needed to evaluate the mechanisms that are involved in these specific bioactive properties.

## 1. Introduction

Platelets derived from the fragmentation of megakaryocytes are anucleate cell fragments that are considered true sentinels of the vascular system [1,2,3,4] because they are abundant in blood [1] and express a variety of functional immunoreceptors. They play an essential role in hemostasis and the wound healing process, which is stimulated by the release of growth factors [5]. Platelets are known to play an important part in the antiinfective host defense [6], in the modulation of the antigen presentation, in the control and modulation of the innate immunity, and in the improvement of adaptive immune responses [1,3]. Due to the production of growth factors, platelets are also important in proliferation, migration, and differentiation of mesenchymal stem cells [4,7]. Platelet-rich plasma (PRP) and platelet-rich fibrin (PRF) are included in the category of platelet concentrates. They are autologous biological products derived from blood and used in regenerative medicine as stimulators of tissue neoangiogenesis [8,9]. PRP is an autologous platelet concentrate that is widely used for multiple medical purposes as a valuable adjuvant for the regeneration of damaged tissues through the concentrated content of growth factors. These include platelet-derived growth factor (PDGF), transforming growth factor beta (TGF-β), vascular endothelial growth factor (VEGF), epidermal growth factor (EGF), and insulin growth factor 1 (IGF-I) [10,11,12,13,14,15,16,17]. The potency and mechanism of action of PRP is highly dependent on the content of functional platelets and the presence and concentration of growth factors, which can activate cells and related signaling pathways [18]. This potential is widely investigated in medicine and veterinary medicine [19,20]. PRF is a platelet concentrate that is formed by a complex network of microfibrin in which platelets and leukocytes are trapped. It holds the ability to release intracellular growth factors, bioactive molecules, and peptides with a prime role in the healing process of hard and soft tissues [21]. These derivatives can be obtained from blood by using low-speed centrifugation, without anticoagulant supplementation [22]. In addition to cytokines, platelets, and leukocytes, PRF also contains circulating stem cells [23]. In addition to these characteristics, the physiological architecture of microfibrin also promotes wound healing and tissue regeneration [15,24]. PRP and PFR can be obtained from blood through various centrifugation procedures, which leads to the obtainment of a plasma fraction with a higher concentration of platelets than that of the circulating blood [25]. The centrifugation speed and time directly influences the type of cells in the concentrate [13]. Thus, based on the low speed of centrifugation, PRF was classified in advanced PRF (A-PRF, centrifugation 14 min at 1300 rpm); advanced PRF plus (advanced PRF+, 8 min of centrifugation at 1300 rpm); injectable PRF (iPRF, 3 min of centrifugation at 700 rpm); injectable PRF obtained from males (iPRF M, 4 min of centrifugation at 700 rpm); injectable PRF plus (iPRF+, 5 min of centrifugation at 700 rpm); and advanced liquid PRF, (A-PRF L, 5 min of centrifugation at 1300 rpm) [5,24,26,27,28]. Although the regenerative potential of platelet concentrates has been widely explored since their introduction into regenerative medicine, especially in oral and maxillofacial surgery, only a few studies have investigated their antimicrobial behavior [5,29]. It is well known that leukocytes are essential elements of the immune system, containing a variety of antimicrobial peptides and enzymes [11,30]. Stimulated leukocytes will degranulate and discharge their contents into phagosomes, followed by destruction of microorganisms by oxidative and nonoxidative reactions [30]. Antimicrobial proteins and peptides from secondary granules are distributed toward the leading edge of the chemotaxing neutrophil, followed by degranulation [31]. The results of multiple studies demonstrate the antibacterial effect of platelet concentrates. Most of the studies indicate that the antibacterial potential is directly correlated with the immune cell concentration [32]. The antimicrobial potential of platelet concentrates is acknowledged, and in some cases it is prominently lower than that of commonly used antibiotics, but there are studies which also show cumulative antimicrobial and antibiofilm effect in the case of combining antibiotics with platelet concentrates [5,33]. Human lactoferrin (Lf) is a bioactive globular protein from the transferrin family, produced by epithelial cells and neutrophils in various mammalian species [34,35,36]. Lf is a first-line defense protein [37] with multiple biological functions, namely antibacterial, antiviral, antitumoral, antifungal, antiinflammatory, and immunomodulatory properties, and even displays analgesic and antioxidant properties [36]. Taking into consideration the global increase in antimicrobial resistance, today’s strategies consist of identifying alternatives for the antimicrobial treatment [38,39]. Within this frame of reference, the purpose of the present study was to assess the cumulative antimicrobial and antibiofilm effect of iPRF functionalized with a multifunctional glycoprotein, Lf. We considered it important to identify nontoxic substances with antimicrobial potential which could represent reliable alternatives in the case of bacterial infections involving resistant and multiresistant microorganisms. We hypothesized that functionalized iPRF owns enhanced bioactive properties against both Gram-positive and Gram-negative biofilm-producing bacteria. The objectives of the study consisted of the assessment of cell proliferation potency, antimicrobial and antibiofilm capacity of iPRF and Lf alone, as well as in combination.

## 2. Results

### 2.1. Proliferation Stimulative Potential of iPRF and Human Lactoferrin

The cell viability of HaCaT cells, cultured with/without iPRF and Lf, and the different combination of these substances was evaluated by using CCK8 assay after 24 and 48 h of incubation in a favorable microclimate (37 °C, 5% CO_2_ and 60–90% relative humidity). CCK assay uses a water-soluble tetrazolium salt, WST-8 [2-(2-methoxy-4-nitrophenyl)-3-(4-nitrophenyl)-5-(2,4disulfophenyl)-2H-tetrazolium, monosodium salt], which is reduced by cellular dehydrogenase of living cells in a water-soluble, orange-colored product [40]. After evaluating the results, no cytotoxic potential was observed for the two substances tested (iPRF and Lf). Twenty-four hours after the treatment with iPRF, no increase in the proliferation degree of treated HaCaT cells was noticed (Figure 1). The average cell viability was 96.33% ± 0.85 in cultures treated with 2% iPRF, 100.73% ± 0.33 for 7% iPRF, and 97.21% ± 1.56 in cultures treated with normal propagation medium supplemented with 10% iPRF. In cultures treated with 7% iPRF, the average cell viability was the highest compared to the other experimental variants, but the differences were not statistically significant compared to the control culture (*p* ≥ 0.05). After 48 h, the average cell viability was 98.81% ± 1.36 in cultures treated with 2% iPRF, 99.88% ± 0.51 for 7% iPRF and 99.05% ± 2.20 in cultures treated with normal propagation medium supplemented with 10% iPRF (Figure 1).

Moreover, after 24 h, in cultures treated with different concentrations of human lactoferrin, the cell proliferation rate indicated an obvious growth compared to control cultures. The average cell viability was 99.34% ± 1.14 for C1 concentration (25 µg/mL), 100.07% ± 0.33 for C2 concentration (50 µg/mL) and 101.68% ± 0.33 for C3 concentration (100 µg/mL). After 48 h from the addition of Lf, the cell proliferation rates indicated an even more pronounced increase compared to the control culture (101.33% ± 0.75 for C1, 98.80% ± 1.53 for C2 and 101.75% ± 0.71 for C3. Therefore, differences were statistically significant, *p* ≤ 0.05 (Figure 2).

The two tested substances were also evaluated in combination for an eventual potentiation of the proliferative capacity. Thus, three different concentrations of Lf and three different concentrations of iPRF were used. These substances were added to the propagation medium after 24 and 48 h, respectively. However, we observed that the combinations significantly increased the cell proliferation rate. After 24 h, the average cell viability of treated cultures (all nine variants) was 102.87% ± 1.00, and the growth tendency was maintained at 48 h, when confluence of the culture and contact inhibition appeared. The highest proliferation rate at 24 h was observed in HaCaT cell cultures treated with 7% iPRF in combination with 50 µg/mL of Lf, where the average cell viability was 105.34 ± 0.47 (the optical density values increased significantly, as compared with untreated control, *p* ≤ 0.05). After 48 h, the average cell viability of treated cultures (all nine variants) was 101.23% ± 0.32, significantly higher than the viability of the control culture. The highest proliferation rate at 48 h was also observed in HaCaT cell cultures treated with 7% iPRF in combination with 50 µg/mL of Lf, where the average cell viability was 102.40 ± 0.80 (Figure 3).

Because the intensity of the chromogenic reaction in the control culture is associated with a cell viability of 100%, all values above this viability, regardless of how low, are considered to have proliferative potential.

### 2.2. Antimicrobial Assay

The antimicrobial potential of a hemocomponent and a glycoprotein from the transferrin family was determined by using the agar well-diffusion assay. Three Gram-positive and two Gram-negative reference bacterial strains were used to assess the antimicrobial property of iPRF, LF and iPRF+Lf. In vitro antimicrobial potential results are presented in Table 1 (mean of growth inhibition zone ± SD). All three concentrations of iPRF showed in vitro antimicrobial potential, but with significant variations depending on the tested bacterial strain. The highest antibacterial activity was identified in S3 against *S. aureus*, where the mean zone of growth inhibition was 14.9 mm ± 0.26, followed by *E. coli* where the mean growth inhibition zone was 13.2 mm ± 0.91. At lower concentrations, a smaller zone of growth inhibition was also observed in *S. aureus*, *S. aureus* MRSA, and *E. coli*. S1 and S2 showed no inhibition against *Pseudomonas aeruginosa* and *Bacillus cereus*, but S3 showed inhibition behavior against both bacterial strains. At the highest concentration of 100 µg/mL, Lf produced a mean of inhibition zone of 12.93 mm ± 0.11 to 16.1 ± 0.1 against Gram-positive bacteria and 12.83 mm ± 0.20 to 18.1 mm ± 0.60 against Gram-negative bacteria. At the lowest concentrations, S1 and S2, Lf showed no activity against *Pseudomonas aeruginosa*, and S1 showed no activity against *Bacillus cereus*. The mixture of iPRF and Lf manifested in vitro antimicrobial efficacity in most concentrations, apart from S1 iPRF +C1 Lf and S1 iPRF+C2 Lf, which did not show any antimicrobial potential against *P. aeruginosa*. The same inefficiency was seen with S1 iPRF+C1 Lf against *B. cereus* strain. S2 iPRF+C3 Lf and S3 iPRF+C3 Lf showed the highest antimicrobial activity against two Gram-positive bacteria (*S. aureus*, *S.aureus* MRSA). The mean inhibition zone diameter was of 19.83 mm ± 0.41 for *S. aureus* MRSA and of 24.53 mm ± 0.40 for *S. aureus,* similar to the positive control gentamicin. The mean was significantly lower against *B. cereus* (14.63 mm ± 0.28). In this strain, however, the degree of inhibition was slightly higher than for gentamicin, where the mean of growth inhibition zone was 14.56 mm ± 0.51. For *E. coli* S2 iPRF+C3, Lf showed the highest antimicrobial potential. The mean growth inhibition zone was of 19.9 mm ± 0.7, higher than the mean produced by gentamicin (18.5 mm ± 0.43). S2 iPRF+C2 Lf showed in vitro antimicrobial potential against *Pseudomonas aeruginosa*, but the mean inhibition zone was significantly lower compared to the standard antibiotic (Table 1). Lf and iPRF showed a significant antimicrobial effect against methicillin-resistant *S. aureus* and methicillin-susceptible *S. aureus* at all studied concentrations. Out of the two agents, Lf exerted a better antimicrobial behavior. Instead, the antimicrobial effect of the two substances on *B. cereus* bore some differences. At concentrations of 2% and 7%, respectively, iPRF did not show any antimicrobial effect. An identical behavior was also observed in the cultures treated with the lowest concentration of lactoferrin. The lack of antimicrobial effect was also seen with the mixture of the lowest concentration of iPRF (2%) and LF (20 µg/mL). In the Gram-negative strains, the tested substances behaved differently. iPRf and Lf alone and in combination showed a bactericidal effect on *E. coli*, and the potential of Lf was higher. Instead, in the case of the *P. aeruginosa*, no inhibitory potential was observed for iPRF 2% and 7%. Similarly, in concentrations of 20 µg/mL and 50 µg/mL, Lf had an identical behavior. When mixed at different concentrations, S1 of iPRf and S1 of Lf did not indicate any antimicrobial behavior. Otherwise, the best mixture turned out to be S2 iPRF with S3 Lf. The antimicrobial potential was the highest against the strain of methicillin-susceptible *S. aureus,* followed by *E. coli* and methicillin-resistant *S. aureus*. The antimicrobial effect was lower against *B. cereus* and *P. aeruginosa*. Several reported studies indicate that the antimicrobial behavior of the two substances has various potential mechanisms. According to the results of Cieślik–Bielecka et al. (2019), [31] the antimicrobial potential of iPRF is probably related to the peptides of leukocytes that have the ability to fuse with the cell membrane of bacteria, followed by the initiation of some metabolic processes, resulting in apoptosis. Antimicrobial peptides also play an important role. Alternatively, the antimicrobial potential of Lf is linked to the iron sequestering property.

The MIC index indicated antimicrobial efficacy of the iPRF and Lf. A more intense antimicrobial potency was noticed against two Gram-positive (*S. aureus* and *S. aureus* MRSA) and one Gram-negative (*E. coli*) bacterial strain (MBC/MIC ≥ 4) (Table 2). The iPRF presented inhibitory potential against methicillin-sensitive *S. aureus* at the concentration of 7% (*v*/*v*) (MIC: 7% *v*/*v*), and at the same concentration acted as MBC (MBC/MIC ≥ 4). iPRF indicated inhibitory and bactericidal activity against methicillin-resistant *S. aureus* and *E. coli* at the concentration of 10% (*v*/*v*) (MIC: 10% *v*/*v*). Lf was found active against methicillin sensitive *S. aureus* for MIC 25 µg/mL, and at the concentration of 50 µg/mL (MBC) no bacterial growth was observed, resulting in MIC index 1. Methicillin-resistant *S. aureus* and the *E. coli* strain exhibited higher MIC and MBC (25 µg/mL and 100 µg/mL, respectively).

The iPRF and lactoferrin mixture, showed a synergistic effect that induced an increase in the antimicrobial potential of the working variants. Bactericidal potential was observed in mixtures with a content of more than 50 µg/mL of lactoferrin and 10% iPRF. No antagonistic effect was identified.

### 2.3. Antibiofilm Assay

iPRF and Lf actively inhibited the biofilm formation in all (three Gram-positive and two Gram-negative) tested bacterial strains after 24 h. The results of our study indicated antibiofilm potential correlated with the concentrations of used substances in all bacterial strains. The average optical densities in the control cultures revealed higher values compared to the bacterial cultures treated with different concentrations of iPRF or Lf (Figure 4a,b) (1.099 ± 0.02 for *S. aureus* MRSA, 1.391 ± 0.15 for *S. aureus*, 1.613 ± 0.29 for *E. coli*, 1.683 ± 0.22 for *Pseudomonas aeruginosa*, and 0.983 ± 0.08 for *Bacillus cereus*).

A significant decrease (*p* ≤ 0.05) of OD was observed in cultures treated with 100 µg/mL Lf, especially in the Gram-positive microorganisms, *S. aureus* MRSA and *S. aureus* (0.757 ± 0.05/0.288 ± 0.06) (Figure 4b).

Similar results were also identified in Gram-negative bacteria (0.483 ± 0.15 for *E. coli* and 0.641 ± 0.05 for *P. aeruginosa*) (Figure 4c). The corresponding concentrations for MBC determined a significant increase in the capacity of bacterial biofilm formation of *S. aureus*, *S. aureus* MRSA, *E. coli*, and *P. aeruginosa*. Mittal et al. (2010) [41] proposed a classification of the specific capacity of biofilm formation, as they consider that values >2.00 correspond to strong biofilm producers, values between 1 and 2 to intermediate biofilm producers, and values <1.00 to weak biofilm producers. The mixture of iPRF and Lf induced a similar behavior compared to the control cultures (Figure 4c–e).

A significant decrease in the biofilm formation capacity was observed depending on the concentrations of Lf. A potentiated effect was seen in most mixtures. The most significant results were noted at increased concentrations of Lf mixed with increased concentrations of iPRF substance (S3 iPRF+C3 Lf; S2 iPRF+C3 Lf; S1 iPRF+C3 Lf). The results are presented in Figure 4d,e.

The antibiofilm potential can be positively correlated with the concentrations of the substances used on all microorganisms. The antibiofilm effect in Gram-positive strains was higher compared to that seen on Gram-negative strains, with the exception of cultures treated with iPRF at the highest concentration of Lf. In this specific case, the optical density was almost equal for both Gram-negative and Gram-positive bacteria. The global analysis of our results suggests that iPRF has antimicrobial potential, but it is low. By comparison, in various concentrations, Lf alone proved a superior antimicrobial effect. The activity of iPRF could thus be potentiated by combining it with different substances, as is the case with Lf. As expected, the highest concentration tested showed the highest antimicrobial and antibiofilm capacity.

## 3. Discussion

Blood derivatives, defined as autologous biological products [42], have been widely used in various fields of medicine and tissue regenerative therapy based on the application of stem cells. These derivatives have three standard characteristics, namely: they have the potential to act as scaffolds, they serve as a source of growth factors through their increased content, and last but not least, they contain living cells [42]. These derivatives are rich in platelets and also contain important growth factors for stimulating the proliferation, migration, and differentiation of progenitor cells [43]. Most of the similar studies carried out previously are focused on the ability to potentiate the differentiation of progenitor cells [22,28,43]. Multiple studies demonstrate the osteogenic potential of these blood derivatives [22]. Generally, they use either stabilized cell lines or primary cell lines isolated from the oral cavity, precisely because their usefulness in dentistry is well known [28]. Obtaining iPRF is an easy procedure; it does not require the use of anticoagulant for preparation and has a higher growth factor content. Its clinical application is not necessarily injectable; more often than not it is used topically (e.g., in superinfected skin wounds, mucosal lesions, together with implants to stimulate regeneration and prevent bacterial superinfections etc.) [16]. This biological therapy with iPRF is considered a simple and cost-effective methodology, used in veterinary medicine to promote tissue healing and regeneration. Derived from the patient’s own blood, it is rich in growth factors and other beneficial substances. Our results demonstrate their effect on keratinocytes focusing only on increasing the proliferation rate (Figure 1). Previous similar studies indicate that platelet concentrates, and especially PRF, also promote keratinocytes’ migration [44]. Through the antiapoptotic effect, Lf can also modulate the capacity of proliferation, adhesion, and cell migration. In addition to the potential to modulate in vitro osteogenesis, the ability to inhibit osteoclastogenesis is also demonstrated [45,46]. Our results regarding the ability to stimulate cell proliferation by Lf are correlative with previously published studies. Through the reasonable combination of these platelet derivatives, a synergistic potential can be obtained, or, in some cases, certain side effects can be reduced. Moreover, the fibrin matrix can serve as a carrier for certain substances [15,29]. Consequently, the combination of these concentrates with different substances could be useful in a clinical approach. Our study implies combinations of these concentrates and a glycoprotein with antimicrobial potential to find a quite feasible alternative to reducing the phenomenon of antibiotic resistance. The major problem in recent decades is considered the appearance of so-called superbugs, i.e., resistant, multiresistant and panresistant bacteria to antimicrobials. Most of these bacteria have an important zoonotic potential [47,48]. The main causes of these pathologies include infections with *Enterococcus faecium, Staphylococcus aureus, Klebsiella pneumoniae, Acinetobacter baumannii, Pseudomonas aeruginosa, Enterobacter* spp. and *Escherichia coli*, together known as *ESKAPE-E*, with relevant intrinsic resistance, but also with the major potential to acquire resistance [49]. The prime characteristic of these microbial agents is that they have the ability to attach to tissues, subsequently forming the biofilm, which makes *ESKAPE-E* infections difficult to prevent and treat [50]. The bacterial biofilm is represented by three-dimensional communities of bacteria, which form both on biotic and abiotic surfaces [51,52]. This capacity is considered a virulence factor, leading to chronic infections. The ability to form biofilms is a common property for both Gram-positive and Gram-negative bacteria. It is an attribute which ensures their survival in adverse conditions. It is a multistage process that is initiated by bacterial adhesion followed by the formation of microcolonies, maturation, and dispersion [53]. In motile bacteria of both Gram types, flagellar protein and secretion of polysaccharides with a role in cell attachment is a significant factor for biofilm formation. Bacterial strains with biofilm formation capacity are 10–1000-fold less susceptible to antibiotics compared to planktonic ones [45,53,54]. In the treatment of various microbial infections, biofilm inhibition is considered as the major target. The potential mechanism of action of iPRF on biofilm-producing bacteria might be correlated to the capacity of degranulated leukocytes to unload the contents into phagosomes, thus leading to the neutralization of bacteria through oxidative and nonoxidative reactions [11]. Another probable antimicrobial and antibiofilm mechanism is related to the permeability proteins it contains, e.g., defensins, heparin-binding protein, cathelicidins and phospholipase A2, and even lactoferrin. These molecules interfere with the metabolic activity of bacterial cells, followed by the occurrence of apoptosis [16]. Thus, combining iPRF with Lf leads to the increase in concentration of Lf, which may underlie the enhanced activity against microorganisms. Currently, there are multiple strategies for the development of such drugs, but different alternative solutions are also considered. Our results demonstrate a clear antimicrobial and antibiofilm potential against methicillin-susceptible *S. aureus*, methicillin-resistant *S. aureus, E. coli, P. aeruginosa,* and *Bacillus cereus*, all with proven zoonotic potential. Due to the emergence and spread of antibiotic resistance, a major global public health issue of the 21st century [55], attempts are being made to identify natural products with antimicrobial potential [52,56]. The hemocomponents, such as PRF, PRP, platelet gel (PG), platelet lysate (PL), and fibrin glue (FG) identified for their regenerative capacities are considered very important products in multiple fields of human and veterinary medicine [6,22,38,55,57]. In the beginning, the PRF was used in the regenerative therapy of soft tissues as membrane, and later the scale of use was extended toward the regenerative therapy of hard tissues. The subsequent development of PRF in an injectable form allowed its use in several fields of medicine [22,58]. In addition to the regenerative potential of these products, the antimicrobial potential is also of major interest. There are multiple studies in this direction, most of them belonging to human medicine. Obviously, the antimicrobial effect of these derivatives is more limited compared to synthetic antimicrobial substances, and therefore their potentiation can confer a higher bioactive potential. Previous studies have emphasized the possible use of platelet-rich fibrin as a carrier matrix. Egle et al. (2022) [29] have used this product together with clindamycin phosphate (a prodrug of clindamycin without antibacterial activity), and the results demonstrated antimicrobial efficacy against *Staphylococcus aureus* and *Staphylococcus epidermidis* [29]. Thus, we considered it appropriate to combine a platelet concentrate, namely iPRF, with a multifunctional iron glycoprotein, lactoferrin, to obtain an increased antimicrobial potential. Synthesized by the epithelial cells and neutrophils, Lf is a multifaceted iron binding bioactive globular protein [36,37,38,59]. In healthy organisms, lactoferrin is predominantly derived from neutrophils and has a concentration of 2−7 × 10^−6^ g/mL [60]. It is considered multifaceted, due to the multiple characteristics it has and because, in addition to the secondary granules of neutrophils, it is present in various exocrine and gastrointestinal secretions [38]. Lf is a first-line defense protein [39] that possesses a number of biological functions, such as antibacterial, antiviral against a wide range of RNA and DNA viruses, antitumor, antifungal, anti-inflammatory, immunomodulatory, analgesic and antioxidant properties. From the multitude of properties of Lf, we chose precisely the antimicrobial potential, with consideration of the current situation in terms of antibiotic resistance. Our results demonstrate this potential, but also their potentiating effect to another natural product with similar properties. Moreover, the present study aimed to provide additional arguments by demonstrating the cell proliferative effect of these functionalized derivatives and also their antibiofilm potential. To evaluate the antimicrobial potential of these substances, multiple studies used similar methods to those applied in our study. The results obtained in previous studies report a bioactive potential for platelet concentrates and for lactoferrin separately [6,9,25,59]. However, as far as we know, there are no studies that try to combine the antimicrobial potential of these two agents. Most of the previously performed studies demonstrate the use of different types of substances, mostly from the range of biomaterials (e.g., hydroxyapatite crystals, nanostructures) that are functionalized with Lf and with promising results in tissue engineering [38,61]. The antimicrobial properties of Lf are the most studied [36,61,62,63]. Several mechanisms are involved in this activity, such as iron chelation, thus depriving microorganisms of this nutrient, or direct interaction with bacterial surface components [36]. The antibacterial mechanism demonstrated in Lf is dissimilar from that of the platelet concentrate, which can therefore demonstrate their cumulative potential, the hypothesis of our study. The bactericidal property of Lf can also occur through direct interaction with bacterial surfaces, with a change in membrane permeability, loss of cell content followed by lysis, with the release of lipopolysaccharide, the component of the outer membrane of Gram-negative bacteria [64,65]. In Gram-positive bacteria, the mechanism of action of Lf is different; cationic residues and hydrophobic residues in the N-terminus disrupt the bacterial membrane [66]. Our results are in agreement with previous in vitro studies performed by Ammons et al. (2013) [45], demonstrating that Lf can inhibit biofilm formation or disrupt existing ones. Scientific data demonstrate weak in vivo bioavailability for Lf; therefore, under the effect of proteolytic enzymes, antimicrobial peptides can appear, the biological activity of which is superior to that of native lactoferrin [66]. However, stabilization can be achieved by incorporating Lf into various collagen-based biomaterials, hydrogels, liposomes, porous microspheres [36], or other biomimetic biomaterials. A very reliable alternative demonstrated by the present study could stand in platelet concentrates derived from blood, which, in addition to stabilizing these glycoproteins, demonstrate a potentiating effect.

## 4. Materials and Methods

### 4.1. Chemicals and Reagents

The iPRF collection tubes were purchased from T-Lab. The CCK-8 assay reagent, Dulbecco’s Modified Eagle Medium (DMEM), antibiotic-antimycotic (100X), MTT ((3-(4,5-dimethylthiazol-2-yl)-2,5-diphenyltetrazolium bromide), crystal violet, and human lactoferrin were obtained from Sigma-Aldrich (St. Louis, MO, USA). Fetal bovine serum was purchased from Gibco Life Technologies (Paisley, UK), Phosphate-buffered saline (PBS) solution was purchased from Lonza Biosciences (Walkersville, MD, USA). The substances for microbiology (nutrient agar, Muller–Hinton broth and Muller–Hinton agar, gentamicin standard disks) were procured from Oxoid (Cambridge, UK).

### 4.2. iPRF and Lf Preparation

Blood samples (*n* = 5) were obtained from a clinically healthy woman (member of our research team after informed consent) and were collected in iPRF collection tubes. Prior to the study, a routine hematological analysis was performed and no modification was found. The experiment was approved by the Institutional Ethical Board of Iuliu Hatieganu University of Medicine and Pharmacy, Cluj-Napoca (No. 281/5 July 2018). The study was performed complying with the Declaration of Helsinki on experimentation involving human subjects.

Processing was performed according to a protocol published by Wang et al. (2017) [67]. Briefly, the samples were centrifuged at 700 rpm for three minutes (60× *g*) at room temperature. The yellow upper layer (1–1.5 mL) was collected in Eppendorf tubes for further use. To prevent the peptide aggregation and release of platelet contents, 0.01% acetic acid was added to the iPRF 68]. Several work variants (iPRF alone, Lf alone, iPRF in combination with Lf) were evaluated in this study. The assessment of cell proliferation potency, antimicrobial and antibiofilm capacity were evaluated at different concentrations of iPRF alone (2%, 7% and 10%, *v*/*v* in culture medium or PBS) and Lf alone (25 µg/mL, 50 µg/mL and 100 µg/mL *v*/*v* in PBS) as well as in combination (2% iPRF + 25 µg/mL Lf; 7% iPRF + 25 µg/mL; 10%PRF + 25 µg/mL; 2% iPRF + 50 µg/mL Lf; 7% iPRF + 50 µg/mL; 10%PRF + 50 µg/mL; 2% iPRF + 100 µg/mL Lf; 7%iPRF + 100 µg/mL; 10%PRF + 100 µg/mL), for a total of 15 working variants. Dilutions were performed at room temperature, under sterile conditions.

### 4.3. Keratinocyte Cell Culture

The HaCaT cell line was kindly provided by the Radiotherapy, Radiobiology and Tumoral Biology Laboratory of the “Ion Chiricuţă” Institute of Oncology Cluj-Napoca, Romania. Cells were cultured in DMEM medium supplemented with 10% fetal bovine serum, 1% antibiotic-antimycotic and maintained at 37 °C in a humidified atmosphere with 5% CO_2_. In order to evaluate the proliferation potential of HaCaT cells, a suspension with a concentration of 1 × 10^5^ cells was seeded in 96-well tissue culture plates containing normal propagation medium (but with reduced content of fetal bovine serum at 5%). After 24 h of incubation, the HaCaT cells were treated with different concentrations of iPRF and Lf and incubated at 37 °C in a humidified atmosphere supplemented with 5% CO2. Negative control was represented by untreated cells (cells maintained in normal propagation medium). Cell viability was measured by using CCK-8 assay following the manufacturer’s protocol. For this purpose, after 24 and 48 h, CCK-8 solution was added to each well, and incubated for an additional 1.5 h. Subsequently, the optical density was determined at 450 nm by using a BioTek Synergy 2 microplate reader (Winooski, VT, USA). The results were expressed as relative viability percentage to the negative control (untreated cells). All experiments were performed in triplicate.

### 4.4. Preparation of Bacterial Suspension—Antimicrobial Assay

The in vitro antimicrobial properties of the iPRF and Lf were assessed by using a Kirby–Bauer well-diffusion assay (according to European Committee on Antimicrobial Susceptibility Testing guidelines, EUCAST) [68]. The iPRF (2, 7, 10% *v*/*v*) and Lf (25, 50, 100 µg/mL) were diluted with PBS. Five bacterial reference strains (*n* = 5), methicillin-susceptible *Staphylococcus aureus* ATCC 25923, methicillin-resistant *Staphylococcus aureus* ATCC 700699 (MRSA), *Escherichia coli* ATCC 25922, *Pseudomonas aeruginosa* ATCC 27853, *Bacillus cereus* ATCC 14579, were used. The overnight bacteria suspensions were prepared according to EUCAST standards, corresponding to McFarland standard 0.5. The Mueller Hinton (MH) agar plates were inoculated by flooding. After plate surface drying, 6-mm-diameter equidistant wells were cut and filled with 60 μL of tested solutions. Standard 6-mm antibiotic discs of Gentamicin (10 µg) were included as reference antimicrobial control. The procedure was performed in independent triplicates. After 24 h of incubation at 37 °C, the mean diameters of growth inhibition zones were evaluated.

### 4.5. Minimum Inhibitory Concentration

The minimum inhibitory concentrations (MICs) were determined by using the microdilution method according to the slightly modified Clinical and Laboratory Standards Institute (CLSI) procedure (2018) [69]. The evaluation was performed by using broth microdilution (twofold dilution) method on 96-well plates, in triplicate. Briefly, 100 µL of Muller–Hinton (MH) broth was added to each well of the 96-well plate, stock solutions of Lf (25–100 µg/mL) were prepared, and 20 µL of bacterial suspension (1.5 × 10^6^ CFU/mL) was added in each well. The iPRF was used at concentrations of 2, 7, and 10% (*v*/*v*). The plates were incubated at 37 °C for 18 h. In order to evaluate bacterial growth/inhibition, after 18 h of incubation, 20 µL of MTT solution 3-(4,5-dimethylthiazol-2-yl)-2,5-diphenyltetrazolium bromide, 1.25 mg/mL) was added to each well. The plate was incubated for 1 h at 37 °C; bacterial growth was indicated by the appearance of purple color and growth inhibition was indicated by a clear/yellow coloration in the well. All tests were performed in triplicate. The MIC was defined as the lowest concentration of substance that completely inhibited the visible bacterial growth in the microdilution wells, compared to control wells (MH broth) [70].

### 4.6. Minimum Bactericidal Concentration

The minimum bactericidal concentration (MBC) value, which represents the lowest concentration at which bacterial growth was completely inhibited, was also assessed. In order to evaluate MBC values, 100 µL of bacterial suspension were collected from the well where no visible bacterial growth was observed. The suspensions were inoculated on MH agar plates and incubated for 18 h at 37 °C. The MIC index was also calculated, based on the MBC/MIC ratio. Thus, an MBC/MIC ration ≤4 was considered bacteriostatic, and an MBC/MIC ration ≥4 was regarded as bactericidal.

### 4.7. Antibiofilm Assay

Antibiofilm formation potential of Lf and iPRF was evaluated in 96-well plates. A total of 300 μL of the overnight bacterial-suspension MH broth (final concentration 1.5 × 10^6^ CFU/mL of *Staphylococcus aureus; Bacillus cereus, Pseudomonas aeruginosa, Escherichia coli*, *and Staphylococcus aureus MRSA*) was added in each well (flat-bottomed 96-well microtitre plates) and incubated at 37 °C for 4 h. Subsequently, the plates were removed from the incubator and the iPRF and Lf solution (15 variants) were added, followed by further incubation at 37 °C for 24 h. Wells with MH broth without testing solutions and MH broth with PBS were considered as controls. After an incubation of 24 h at 37 °C, the supernatant was discarded in order to eliminate the floating cells, and each well was washed with sterile PBS. Later, the plates were air-dried for 30 min and the resulting biofilm was stained with 0.1% aqueous solution of crystal violet for 30 min. The plates were washed three time with sterile PBS in order to remove the excess dye, followed by solubilization of crystal violet with 200 µL 70% ethanol solution. The plates were incubated for 15 min at room temperature. After incubation, 200 µL of dissolved crystal violet solution was transferred to a new 96-well plate and the optical density was determined at 450 nm by using a BioTek Synergy 2 microplate reader (Winooski, VT, USA). The results were calculated by using the following formula: biofilm inhibition = OD of the attached stained bacteria—OD blank (bacteria free medium)/OD of bacterial suspension maintained in MH broth [71].

### 4.8. Statistical Analysis

The one-way ANOVA and *t*-test (GraphPad Prism 8) were used for statistical analysis. The results were expressed as mean ± standard deviation (SD); *p* ≤ 0.05 was considered statistically significant.

## 5. Conclusions

Our findings underline significant antimicrobial and antibiofilm capacity for iPRF, Lf, and their combinations. The lack of cytotoxicity and, moreover, the ability to potentiate cell proliferation brings additional arguments to strongly consider not only the substances themselves, but especially the specific combination between them, for further investigation. According to our knowledge, this is the first study that combines iPRF and LF in order to stimulate the increase of HaCaT cells proliferation and augment the antimicrobial and antibiofilm activity against Gram-positive and Gram-negative bacteria. The functionalization of iPRF with lactoferrin demonstrates a cell proliferative potential and the distinct antimicrobial and antibiofilm behavior against methicillin-susceptible *S. aureus* and methicillin-resistant *S. aureus*, *E. coli*, *P. aeruginosa*, and *Bacillus cereus*. Nevertheless, further studies are needed to evaluate the specific mechanisms that are involved in the possible potentiation of these two substances. These results can easily be extrapolated to the veterinary field and offer support to clinicians in their effort to either prevent infectious pathologies, or to heal and regenerate affected tissues.

## Figures and Tables

**Figure 1 molecules-28-01943-f001:**
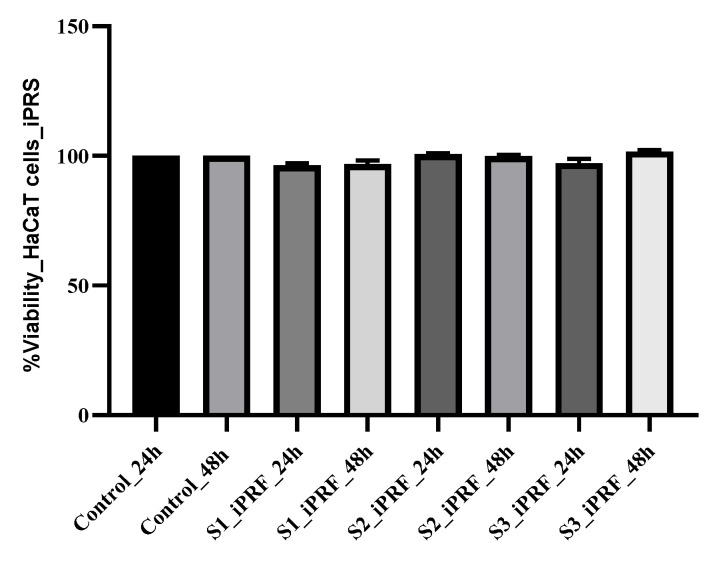
HaCaT cell proliferation kinetics after 24 and 48 h of treatment with different concentrations of iPRF (S1, normal propagation medium supplemented with 2% of iPRF; S2, normal propagation medium supplemented with 7% of iPRF; S3, normal propagation medium supplemented with 10% of iPRF). The control was represented by HaCaT cells maintained in normal propagation medium (DMEM + 10%FCS + 1%AA). The results were compared to control (untreated cells cultures) and expressed as the average cell viability obtained after testing in triplicate.

**Figure 2 molecules-28-01943-f002:**
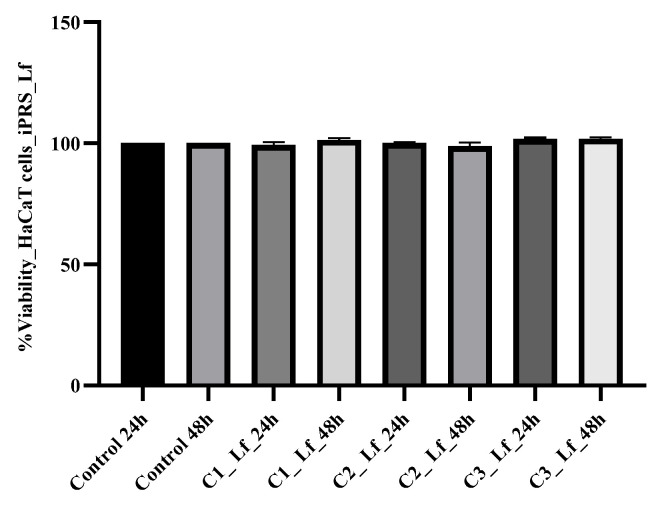
HaCaT cells proliferation kinetics after 24 and 48 h of treatment with different concentrations of Lf (C1 25 µg/mL), C2 50 µg/mL and C3 100 µg/mL. Lf, human lactoferrin dissolved in PBS. The results were compared to control (untreated cells cultures) and expressed as the average cell viability obtained after testing in triplicate.

**Figure 3 molecules-28-01943-f003:**
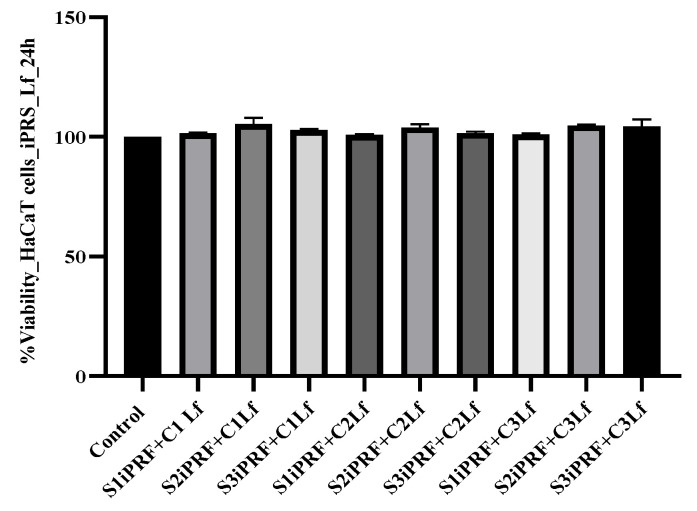
HaCaT cells proliferation kinetics after 24 h of treatment with the iPRF and Lf mixture in several work variants (S1iPRF+C1 Lf, 2% iPRF and 25 µg/mL Lf; S2PRF+C1Lf, 7% iPRF and 25 µg/mL Lf; S3iPRF+C1 Lf, 10% iPRF and 25 µg/mL Lf; S1iPRF+C2 Lf, 2% iPRF and 50 µg/mL Lf; S2PRF+C2Lf—7% iPRF and 50 µg/mL Lf; S3iPRF+C2 Lf, 10% iPRF and 50 µg/mL Lf; S1iPRF+C3 Lf, 2% iPRF and 75 µg/mL Lf; S2PRF+C3Lf, 7% iPRF and 75 µg/mL Lf; S3iPRF+C3 Lf, 10% iPRF and 75 µg/mL Lf). The results were compared to control (untreated cells cultures) and expressed as the average cell viability obtained after testing in triplicate.

**Figure 4 molecules-28-01943-f004:**
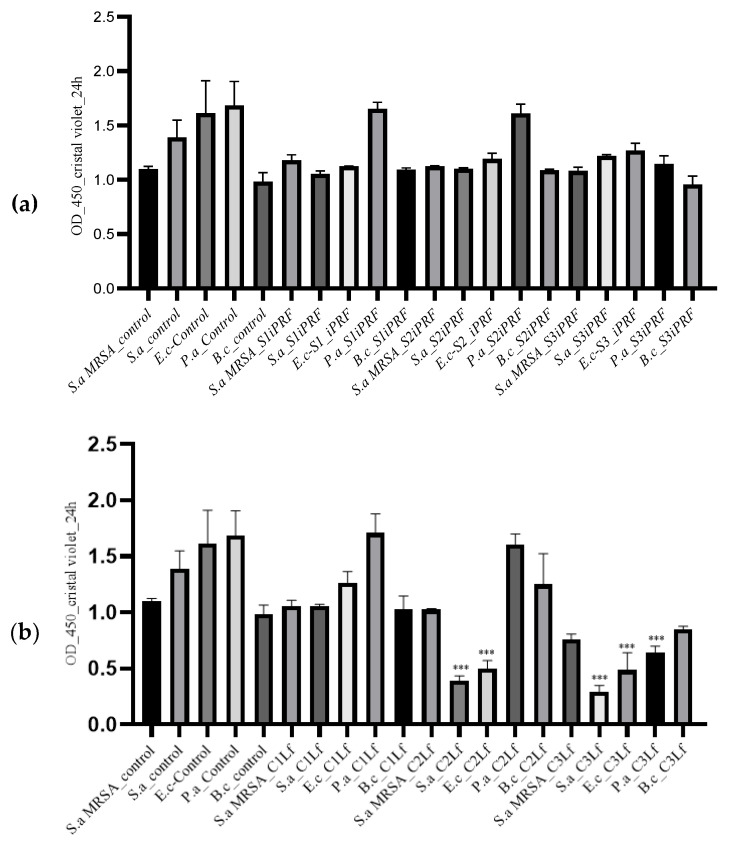
(**a**) Activity of biofilm prevention of iPRF against Gram-positive and Gram-negative bacteria (Sa MRSA, *S. aureus* MRSA; Sa, *S. aureus*; Ec, *E. coli*, Pa, *P. aeruginosa*, Bc, *Bacillus cereus*; control cultures and iPRF.) Control cultures and bacterial strains treated with iPRF in three different concentrations (S1, S2, S3) (*v*/*v*); Biofilm formation using microtiter method and evaluation after crystal violet staining. Data are mean ± SD (*n* = 3). The results are reported to control. Statistical analyses were performed by one-way ANOVA and *t*-test, and *p* ≤ 0.05 is considered statistically significant. (**b**) Activity of biofilm prevention of iPRF against Gram-positive and Gram-negative bacteria (Sa MRSA, *S. aureus* MRSA; Sa, *S. aureus*; Ec, *E. coli*; Pa, *P. aeruginosa*; Bc, *Bacillus cereus*; control cultures and Lf). Control cultures and bacterial strains treated with Lf in three different concentrations (C1, C2, C3) (µg/mL); biofilm formation using microtiter method and evaluation after crystal violet staining. Data are mean ± SD (*n* = 3). The results are reported to control. Statistical analyses were performed by one-way ANOVA and *t*-test, and *p* ≤ 0.05 is considered statistically significant; * *p* ≤ 0.05, ** *p* < 0.01, *** *p* < 0.001. (**c**) Activity of biofilm prevention of iPRF against Gram-positive and Gram-negative bacteria (Sa MRSA, *S. aureus* MRSA; Sa, *S. aureus*; Ec, *E. coli*; Pa, *P. aeruginosa*; Bc, *Bacillus cereus*; control cultures and Lf). Control cultures and bacterial strains treated with different combinations of iPRF (*v*/*v*)/Lf (µg/mL) in three different concentrations. Biofilm formation using microtiter method and evaluation after crystal violet staining. Data are mean ± SD (*n* = 3). The results are reported to control. Statistical analyses were performed by one-way ANOVA ad *t*-test, and *p* ≤ 0.05 is considered statistically significant; * *p* ≤ 0.05, ** *p* < 0.01, *** *p* < 0.001; (**d**,**e**) Activity of biofilm prevention of iPRF and Lf against Gram-positive and Gram-negative bacteria (Sa MRSA, *S. aureus* MRSA; Sa, *S. aureus*; Ec, *E. coli*; Pa, *P. aeruginosa*; Bc, *Bacillus cereus*). Control cultures and bacterial strains treated with different combinations of iPRF (*v*/*v*)/Lf (µg/mL) in three different concentrations. Biofilm formation using microtiter method and evaluation after crystal violet staining. Data are mean ± SD (*n* = 3). The results are reported to control. Statistical analyses were performed by one-way ANOVA ad *t*-test; *p* ≤ 0.05 is considered statistically significant; * *p* ≤ 0.05, ** *p* < 0.01, *** *p* < 0.001.

**Table 1 molecules-28-01943-t001:** Mean values of growth inhibition zones of iPRF, Lf and reference drug against tested bacterial strains.

Tested Solutions	*Staphylococcus* *aureus MRSA* (mm)	*Staphylococcus aureus* (mm)	*Escherichia coli* (mm)	*Pseudomonas aeruginosa* (mm)	*Bacillus cereus* (mm)
iPRF2%	12.40 ± 0.55	14.16 ± 0.37	10.53 ± 0.64	na	na
iPRF 7%	13.03 ± 0.61	14.10 ± 0.65	11.10 ± 1.15	na	na
iPRF 10%	11.50 ± 0.10	14.90 ± 0.26	13.20 ± 0.91	8.53 ± 0.51	12.20 ± 1.05
25 µg/mL Lf	11.90 ± 0.10	13.36 ± 0.40	12.53 ± 0.15	na	na
50 µg/mL Lf	13.73 ± 0.64	14.43 ± 0.40	12.50 ± 0.43	na	11.90 ± 0.83
100 µg/mL Lf	15.43 ± 0.68	16.10 ± 0.10	18.10 ± 0.60	12.83 ± 0.20	12.93 ± 0.11
S1 iPRF+C1 Lf	13.26 ± 0.95	18.46 ± 0.37	13.53 ± 0.61	na	na
S2 iPRF+C1 Lf	13.26 ± 1.34	17.90 ± 0.52	13.66 ± 0.23	na	12.06 ± 0.56
S3 iPRF+C1 Lf	13.33 ± 0.80	18.73 ± 0.64	14.40 ± 0.36	9.90 ± 1.65	13.30 ± 0.34
S1 iPRF+C2 Lf	15.23 ± 1.80	20.20 ± 0.43	14.63 ± 0.30	12.80 ± 0.26	13.83 ± 0.23
S2 iPRF+C2 Lf	13.73 ± 0.70	21.96 ± 0.15	15.03 ± 0.35	14.30 ± 0.75	12.90 ± 0.10
S3 iPRF+C2 Lf	13.60 ± 0.50	22.30 ± 0.30	15.86 ± 0.15	12.40 ± 0.60	13.63 ± 0.30
S1 iPRF+C3 Lf	15.60 ± 0.88	23.96 ± 0.05	18.46 ± 0.35	13.63 ± 0.58	15.06 ± 0.51
S2 iPRF+C3 Lf	18.16 ± 0.40	24.53 ± 0.40	19.90 ± 0.70	12.53 ± 0.41	15.30 ± 0.26
S3 iPRF+C3 Lf	19.83 ± 0.41	23.73 ± 0.66	17.56 ± 0.70	12.50 ± 0.10	14.63 ± 0.28
Gentamicin	25.66 ± 1.44	27.50 ± 1.32	18.50 ± 0.43	17.00 ± 0.30	14.56 ± 0.51

The mean zone of inhibition was determined from three independent results (*n* = 3) mean ± SD; na, not active; SD, standard error; iPRF, injectable platelet fibrin; Lf, lactoferin; S1, solution 1 iPRF; S2, solution 3 iPRF; S3, solution 3 iPRF; C1, concentration 1 for Lf; C2, concentration 2 for Lf; C3, concentration 3 for Lf; Gentamicin 10 µg, na—not active.

**Table 2 molecules-28-01943-t002:** Microdilution assay MIC, MBC, MIC index values of iPRF and Lf against tested bacteria strains.

MIC Index MBC/MIC	*Staphylococcus aureus*	*Staphylococcus aureus* MRSA	*Escherichia coli*	*Pseudomonas aeruginosa*	*Bacillus cereus*
iPRF % *v*/*v*	1 7/7	1 10/10	1 10/10	na	na
Lf µg/mL	2 25/50	1 50/50	1 100/100	na	na

na—not active.

## Data Availability

Not applicable.

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
