# Peer review of "Enhanced Bioactive Potential of Functionalized Injectable Platelet-Rich Plasma"

_molecules, 2023, doi:10.3390/molecules28041943_

Round 1
Reviewer 1 Report
This manuscript discuss about the Enhanced bioactive potential of functionalized injectable platelet rich plasma . An interesting knowledge has been reported, however the following comments should be addressed before acceptance
Comments
- The novelty of the manuscript must be better emphasizsed
- Abstract section - Suggested to add some qualitative results, thus could enhance the readability
-in introduction section; a clear aim/objective of the study should be given at the end of the introduction section
-author have followed Kirby Bauer disc-diffusion assay for antimicrobial analysis- suggested to give images of plates, if it is possible
-How does concentration affects the biofilm activity
-Enhance the quality of the images
Compare antimicrobial results between both Gram positive and Gram negative bacterial strains, which is more inhibited, and discuss why?
Suggested to revise the conclusion part, add some interesting results/findings, and add how this research helpful to lead better this research area
After addressing all the comments, this manuscript can be acceptable for further progress
Author Response
This manuscript discuss about the Enhanced bioactive potential of functionalized injectable platelet rich plasma. An interesting knowledge has been reported, however the following comments should be addressed before acceptance
Authors generic response: The authors thank the reviewer for all the suggestions meat to improve the present form of the paper. We tried to follow every one of them and we hope that the modified from fulfills all the necessary demands of the journal. All the suggested changes were added afterwards, according to the suggestions of the reviewer.
Reviewer 1 Comments
- The novelty of the manuscript must be better emphasizsed
Authors response 1. According to our knowledge, this is the first study that combines iPRF and LF in order to stimulate the increase of HaCaT cells proliferation, and augment the antimicrobial and antibiofilm activity against Gram-positive and Gram-negative bacteria. Thus, the functionalization of iPRF with lactoferrin demonstrates a cell proliferative potential and a distinct antimicrobial and antibiofilm behavior against methicillin-susceptible S. aureus, methicillin resistant S. aureus, E. coli, P. aeruginosa, and Bacillus cereus.
Please find this added to Conclusions.
Abstract section - Suggested to add some qualitative results, thus could enhance the readability
Authors response 2. Thank you very much for the suggestions. We have also added qualitative data. They can be seen inserted and marked in red in the original text.
Abstract: Injectable platelet rich fibrin (iPRF) is a frequently used platelet concentrate for various medical purposes both in veterinary and human medicine due to the regenerative potential of hard and soft tissues, but also due to its antimicrobial effectiveness. This in vitro study was carried out to assess the cumulative antimicrobial and antibiofilm effect of iPRF functionalized with a multifunctional glycoprotein, human lactoferrin. Thus, the ability to potentiate cell proliferation was tested on keratinocytes and evaluated by the CCK8 test. The combinations of iPRF and Lf induced an increase in the proliferation rate after 24 hours. The average cell viability of treated cultures (all 9 variants) was 102.87%±1.00, and the growth tendency was maintained even at 48 hours. The highest proliferation rate was observed in cultures treated with 7% iPRF in combination with 50 µg/ml of Lf, with an average viability of 102.40%±0.80. The antibacterial and antibiofilm activity of iPRF, of human lactoferrin and their combination were tested by agar-well diffusion (Kirby–Bauer assay), broth microdilution and Crystal violet assay against five reference bacterial strains. iPRF showed antimicrobial and antibiofilm potential, but with variations depending on the tested bacterial strain. The global analysis of the results indicates an increased antimicrobial potential at the highest concentration of Lf mixed with iPRF.The study findings confirmed the hypothesized enhanced bioactive properties of functionalized iPRF against both Gram-positive and Gram-negative biofilm-producing bacteria. These findings could be further applied, but additional studies are needed to evaluate the mechanisms that are involved in these specific bioactive properties.
-in introduction section; a clear aim/objective of the study should be given at the end of the introduction section
Authors response 3. We hypothesized that functionalized iPRF owns enhanced bioactive properties against both Gram-positive and Gram-negative biofilm-producing bacteria. The proliferative potential was evaluated in HaCaT cells and the potential antimicrobial and antibiofilm effect against three Gram-positive and two Gram-negative bacterial strains.
Please find this now inserted in the paper.
-author have followed Kirby Bauer disc-diffusion assay for antimicrobial analysis- suggested to give images of plates, if it is possible
Authors response 4. We consider the pictures taken during the study have poor quality and therefore were not added to the manuscript. Please observe some of these pictures.
-How does concentration affects the biofilm activity
Authors response 5. The antibiofilm potential can be positively correlated with the concentrations of the substances used on all the microorganisms. The antibiofilm effect in Gram-positive strains was higher compared to that seen on Gram-negative strains, with the exception of cultures treated with iPRF at the highest concentration of Lf. In this specific case, the optical density was almost equal for both Gram-negative and Gram-positive bacteria. The global analysis of our results suggests that iPRF has antimicrobial potential, but it is low. By comparison, in various concentrations, Lf alone proved a superior antimicrobial effect. The activity of iPRF could thus be potentiated by combining it with different substances, as is the case with Lf. As expected, the highest concentration tested showed the highest antimicrobial and antibiofilm capacity.
The bacterial biofilm is represented by three-dimensional communities of bacteria, which form both on biotic and abiotic surfaces [57, 58]. This capacity is considered a virulence factor, leading to chronic infections. The ability to form biofilms is a common property for both gram-positive and gram-negative bacteria. It is an attribute which ensures their survival in adverse conditions. It is a multistage process that is initiated by bacterial adhesion followed by the formation of microcolonies, maturation and dispersion (Ruhal and Kataria, 2021). In motile bacteria of both Gram types, flagellar protein and secretion of polysaccharides with a role in cell attachment is a significant factor for biofilm formation. Bacterial strains with biofilm formation capacity are 10–1000 fold less susceptible to antibiotics compared to planktonic ones [51, 59]. In the treatment of various microbial infections, biofilm inhibition is considered as the major target. The potential mechanism of action of iPRF on biofilm-producing bacteria might be correlated with the capacity of degranulated leukocytes, which have the potential to unload the contents into phagosomes, thus leading to the neutralization of bacteria through oxidative and non-oxidative reactions (Jasmine et al., 2020). Another probable antimicrobial and antibiofilm mechanism is related to the permeability proteins it contains, e.g. defensins, heparin-binding protein, cathelicidins and phospholipase A2 and even lactoferrin. These molecules interfere with the metabolic activity of bacterial cells, followed by the occurrence of apoptosis (Jasmine et al., 2020). Thus, combining iPRF with Lf leads to the increase in concentration of Lf, which may underlie the enhanced activity against microorganisms.
We have added this fragment to Discussions section.
Rohit Ruhal, Rashmi Kataria,Biofilm patterns in gram-positive and gram-negative bacteria, Microbiological Research, Volume 251, 2021,126829, ISSN 0944-5013.
Jasmine, S., A, T., Janarthanan, K., Krishnamoorthy, R., & Alshatwi, A. A. (2020). Antimicrobial and antibiofilm potential of injectable platelet rich fibrin-a second-generation platelet concentrate-against biofilm producing oral staphylococcus isolates. Saudi journal of biological sciences, 27(1), 41–46. https://doi.org/10.1016/j.sjbs.2019.04.012
-Enhance the quality of the images
Authors response 6. The quality of the images has been improved.
Compare antimicrobial results between both Gram positive and Gram negative bacterial strains, which is more inhibited, and discuss why?
Authors response 7. Lf and iPRF showed a significant antimicrobial effect against methicillin resistant S. aureus and methicillin-susceptible S. aureus at all studied concentrations. Out of the two agents, Lf exerted a better antimicrobial behavior. Instead, the antimicrobial effect of the two substances on B. cereus bore some differences. At concentrations of 2% and 7%, respectively, iPRF did not show any antimicrobial effect. An identical behavior was also observed in the cultures treated with the lowest concentration of lactoferrin. At the lowest concentration of iPRF (2%) and LF (20 µg/ml), no antimicrobial effect was found. In the Gram-negative strains, the tested substances behaved differently. iPRf and Lf alone and in combination showed an bactericidal effect on E. coli, and the potential of Lf was higher. Instead, in the case of the P. aeruginosa, no inhibitory potential was observed for iPRF 2% and 7%. Similarly, in concentrations of 20 µg/ml and 50 µg/ml, Lf had an identical behavior. When mixed at different concentrations, S1 of iPRf and S1 of Lf did not indicate any antimicrobial behavior. Otherwise, the best mixture turned out to be S2 iPRF with S3 Lf. The antimicrobial potential was the highest against the strain of methicillin-susceptible S. aureus, followed by E. coli and methicillin resistant S. aureus. The antimicrobial effect was lower against B. cereus and P. aeruginosa. Several reported studies indicate that the antimicrobial behavior of the two substances has various potential mechanisms. According to the results of Cieślik-Bielecka et col. (2019), the antimicrobial potential of iPRF is probably related to the peptides of leukocytes that have the ability to fuse with the cell membrane of bacteria, followed by the initiation of some metabolic processes, resulting in apoptosis. Antimicrobial peptides also play an important role. Alternatively, the antimicrobial potential of Lf is linked to the iron sequestering property.
Please find this inserted in the Subsection 2.2 Antimicrobial assay
Suggested to revise the conclusion part, add some interesting results/findings, and add how this research helpful to lead better this research area.
Authors response 8.
We have added a few lines to Conclusions and hope to have improved it.
“Our findings underline significant antimicrobial and antibiofilm capacity for iPRF, Lf and their combinations. The lack of cytotoxicity and, moreover, the ability to potentiate cell proliferation brings additional arguments to strongly consider not only the substances themselves, but especially the specific combination between them, for further investigation. According to our knowledge, this is the first study that-which combines iPRF and LF in order to stimulate the increase of HaCaT cells prolifer-ation, and augment the antimicrobial and antibiofilm activity against Gram-positive and Gram-negative bacteria. Thus, Tthe functionalization of iPRF with lactoferrin demonstrates a cell proliferative potential and a the distinct antimicrobial and antibio-film behavior against methicillin-susceptible S. aureus, methicillin resistant S. aureus, E. coli, P. aeruginosa, and Bacillus cereus. Nevertheless, further studies are needed to eval-uate the specific mechanisms that are involved in the possible potentiation of these two substances.
All results can easily be extrapolated to the veterinary field and be useful in veterinary stomatology as well as prevention and control infectious diseases in animals.
After addressing all the comments, this manuscript can be acceptable for further progress
On behalf of the authors of the manuscript, I thank you for the consideration to our work and the Reviewer 1 for the suggestions that helped to significantly improve the quality of the manuscript.
Sincerely yours,
Emoke Pall

Reviewer 2 Report
Dear authors please forgive my language, I will go into details directly to save time. Overall the results have enough importance to published but presentation of the results and discussion of the results do not have enough scientific soundness. This should be improved and then can be published in my opinion. There is huge pile of data need to be digested and reading it as a whole is difficult.
The submitted study uses platelet concentrates for medical purposes including their antimicrobial activity. In vitro experiments were performed using platelet rich fibrin solutions along with lactoferrin antimicrobial peptides. Antibacterial and antibiofilm activities were monitored.
Comments on article:
1. The paragraphs are too long to follow. They need to be break into smaller pieces.
2. Abstract is missing the most important results of the study. It says “The study findings confirmed the hypothesized enhanced bioactive properties of functionalized iPRF against both Gram-positive and Gram-negative biofilm-producing bacteria”. How this is different from the literature or how it adds up to the literature. This should mentioned briefly in the abstract with clearly represented results.
1. Materials and methods section give sufficient details of the study.
2. Introduction, results and discussion sections must be divided into pieces. It is not easy to follow for the reader. There is no use of paragraphs under the same title. Please break those big paragraphs into pieces. Especially in the results section you should give the related results right after you talked about them. This will be a lot easier. In the manuscript results are piled up in a big paragraph and graphs were added right after each other. This causes lots of difficulties during reading and makes it really challenging to understand and follow. It is really complicated to follow those numbers in just one paragraph.
3. In line 80, when you say cumulative effect, do you mean antimicrobial or any other effect? Please explain.
4. Importance of the study was not given with enough details, It is not easy to find the hypothesis.
5. In figure 1 graph shows a line but the values are from separate biological tests. Columns must be used since them samples are not a part of a continuum in an experiment. Same thing goes for the figure 2.
6. I suggest to use 0% as the baseline on X axis in the graphs. It seems like there is a big difference at a first glance.
7. Even though your statistical analyses show significant difference, only 2% max difference seems not that much. Address how this little difference becomes so important as a scientific result. In other words, 1-2% difference must be justified as important.
8. Please show statistical data between the results on the graph. You can check literature for it. When you put ** signs you also need to show which value you are comparing. If this comparison is only made by the control say so.
9. In line 159 E. coli should be italicized. Leave a gap after the dot. Please use a right nomenclature for the species names. Don’t use the abbreviated species names before giving the full name first. Please correct these through the manuscript.
10. In line 184 “activite” is not English.
11. Table 1 should include units (mm or cm?). It is not clear. Reader should be able to extract this info from the table itself without reading the main text.
12. In line 157, it says “highest activity” but S1, S3 on the table show higher values than this? Please clarify. 14.9±0.26 is the result of iPRF% not S2. Please clarify.
13. The result section should be written again with more clear statements. It is to difficult to follow. You can divide table 1 into smaller pieces, so it will be easy to follow.
14. Figure 5 shows statistical data as ** or ***, but it is not clear which of the two data points are being compared. You can connect the compared point with a line on the bars and put these statistical data (**) on that line. If these are compared to control you have to mention which control it is.
15. The graphs are too small, based on the guideline for the authors of this journal there is no page limitation. Then don’t put all your data in tiny graphs.
16. Separate the figures into smaller figures and discuss each in a separate paragraph. It is too difficult to follow.
17. The statements in conclusion section is too general, please try to make it more specific for your results.
18. The manuscript is submitted into “Medicinal Chemistry” section / “Veterinary Drugs” special issue. However it does not have any chemical analyses. Therefore I suggest it to be submitted under Natural Products section. Veterinary side of the results were not discussed properly. More literature comparison can be made to make the manuscript a good fit for the Veterinary Drugs issue. Manuscript does not talk about any animal diseases. On the other side it uses human cell line (HaCaT cells). Also the blood samples used in the study were taken from human. This is confusing please clarify or add more details on veterinary side.
Author Response
Dear authors please forgive my language, I will go into details directly to save time. Overall the results have enough importance to published but presentation of the results and discussion of the results do not have enough scientific soundness. This should be improved and then can be published in my opinion. There is huge pile of data need to be digested and reading it as a whole is difficult.
The submitted study uses platelet concentrates for medical purposes including their antimicrobial activity. In vitro experiments were performed using platelet rich fibrin solutions along with lactoferrin antimicrobial peptides. Antibacterial and antibiofilm activities were monitored.
Authors generic response: The authors thank the reviewer for all the suggestions meant to improve the present form of the paper. We tried to follow every one of them and we hope that the modified form fulfills all the necessary demands of the journal. All the suggested changes were added afterwards, according to the suggestions of the reviewer.
Comments on article:
- The paragraphs are too long to follow. They need to be break into smaller pieces.
Authors response 1. Thank you very much for the suggestion, we reformulated the long sentences and separated them.
- Abstract is missing the most important results of the study. It says “The study findings confirmed the hypothesized enhanced bioactive properties of functionalized iPRF against both Gram-positive and Gram-negative biofilm-producing bacteria”. How this is different from the literature or how it adds up to the literature. This should mentioned briefly in the abstract with clearly represented results.
Authors response 2. We have modified the summary and completed it with the data you suggested. Thank you.
- Materials and methods section give sufficient details of the study.
Introduction, results and discussion sections must be divided into pieces. It is not easy to follow for the reader. There is no use of paragraphs under the same title. Please break those big paragraphs into pieces. Especially in the results section you should give the related results right after you talked about them. This will be a lot easier. In the manuscript results are piled up in a big paragraph and graphs were added right after each other. This causes lots of difficulties during reading and makes it really challenging to understand and follow. It is really complicated to follow those numbers in just one paragraph.
Authors response 3. We thank you for the insightful suggestion. We have organized the article in accordance with the standards of the journal indicated in the guidelines for authors. Consulting other publications that appeared in Molecules, we did not find articles where the Introduction and Discussions are divided into pieces. The results are still divided into subsections according to the organizational chart of the study, but we have divided and inserted the charts in between fragments, hoping to make the article easier to follow.
- In line 80, when you say cumulative effect, do you mean antimicrobial or any other effect? Please explain.
Authors response 4. We apologize for the negligence. We corrected, with “cumulative antimicrobial and antibiofilm effect”.
- Importance of the study was not given with enough details, It is not easy to find the hypothesis.
Authors response 5. We considered it important to identify non-toxic substances with antimicrobial potential which could represent reliable alternatives in the case of bacterial infections involving resistant and multi-resistant strains to antibiotics. We hypothesized that functionalized iPRF owns enhanced bioactive properties against both Gram-positive and Gram-negative biofilm-producing bacteria.
Please find this fragment added in the paper at the end of the Introduction.
- In figure 1 graph shows a line but the values are from separate biological tests. Columns must be used since them samples are not a part of a continuum in an experiment. Same thing goes for the figure 2.
Authors response 6. We have now changed the style of the figures. (In figures, 2, 3 and 4, the respective line indicated the average cell viability at 24 and 48 hours, respectively, after the addition of iPRF, Lf and the mixture of iPRF and Lf at different concentrations to the culture medium.)
- I suggest to use 0% as the baseline on X axis in the graphs. It seems like there is a big difference at a first glance.
Authors response 7. Since the average cell viability is over 90%, we cannot select 0% because the statistical program we used does not allow us.
- Even though your statistical analyses show significant difference, only 2% max difference seems not that much. Address how this little difference becomes so important as a scientific result. In other words, 1-2% difference must be justified as important.
Authors response 8. In the proliferation tests on cell cultures, the results are reported against the control, which is represented by a culture maintained in a specific propagation medium. The intensity of the chromogenic reaction in the control culture is associated with a cell viability of 100%. By applying the calculation formula, all values above this viability, regardless how low, are considered to have proliferative potential.
Please find this next fragment added in the paper at the end of the subsection Proliferation stimulative potential of iPRF and human lactoferrin.
“Since the intensity of the chromogenic reaction in the control culture is associated with a cell viability of 100%, all values above this viability, regardless how low, are considered to have proliferative potential.”
- Please show statistical data between the results on the graph. You can check literature for it. When you put ** signs you also need to show which value you are comparing. If this comparison is only made by the control say so.
Authors response 9. Thank you very much for the clarification. We corrected according to your suggestions. We must specify that the comparison was made with the control results.
- In line 159 E. coli should be italicized. Leave a gap after the dot. Please use a right nomenclature for the species names. Don’t use the abbreviated species names before giving the full name first. Please correct these through the manuscript.
Authors response 10. We have made the suggested corrections.
- In line 184 “activite” is not English.
Authors response 11. We have made the correction.
- Table 1 should include units (mm or cm?). It is not clear. Reader should be able to extract this info from the table itself without reading the main text.
Authors response 12. The values are in mm. We have added in the table under each tested bacterial strain.
- In line 157, it says “highest activity” but S1, S3 on the table show higher values than this? Please clarify. 14.9±0.26 is the result of iPRF% not S2. Please clarify.
Authors response 13. Thank you very much, we corrected it, it was actually about the S3.
- The result section should be written again with more clear statements. It is to difficult to follow. You can divide table 1 into smaller pieces, so it will be easy to follow.
Authors response 14. The results section has been modified in a significant extent. We did not divide the table by categories of substances, because we considered it more suggestive and the results easier to compare as they are presented now.
- Figure 5 shows statistical data as ** or ***, but it is not clear which of the two data points are being compared. You can connect the compared point with a line on the bars and put these statistical data (**) on that line. If these are compared to control you have to mention which control it is.
Authors response 15. The comparison was made to control. It is now mentioned in the paper and under each figure.
- The graphs are too small, based on the guideline for the authors of this journal there is no page limitation. Then don’t put all your data in tiny graphs. Separate the figures into smaller figures and discuss each in a separate paragraph. It is too difficult to follow.
Author response 16. Figure 5 has been split and inserted into the text.
- The statements in conclusion section is too general, please try to make it more specific for your results.
Author response 17. Thank you very much for the suggestion. Conclusions have been completed.
- The manuscript is submitted into “Medicinal Chemistry” section / “Veterinary Drugs” special issue. However it does not have any chemical analyses. Therefore I suggest it to be submitted under Natural Products section. Veterinary side of the results were not discussed properly. More literature comparison can be made to make the manuscript a good fit for the Veterinary Drugs issue. Manuscript does not talk about any animal diseases. On the other side it uses human cell line (HaCaT cells). Also the blood samples used in the study were taken from human. This is confusing please clarify or add more details on veterinary side.
Author response 18. I chose to send our manuscript for publication in Molecules, special issue Veterinary Drugs, because I am a veterinarian involved in translational research for many years and the current manuscript fits the scope of this special issue. Since I was invited by the guest editors to submit my research here, I considered Molecules as an important journal, with suitable readership and outreach, and decided that our work could reach a adequate audience if published here. These results can easily be extrapolated to the veterinary field and offer support to clinicians in their effort to either prevent infectious pathologies, or to heal and regenerate affected tissues.
On behalf of the authors of the manuscript, I thank you for the consideration to our work and the Reviewer 2 for the suggestions that helped to significantly improve the quality of the manuscript.
Sincerely yours,
Emoke Pall

Reviewer 3 Report
Figure 1 -4 Bar chart should be used instead of joining the points of with relevance. Any statistical significance should be indicated on the bars too. Nature of error bar and number of replicate should be shown as well.
Table 1 Unit of the number not indicated throughout the Table. The number of decimal places is not consistent in the Table. Any statistical analysis has been done?
Page 11. 4.2 Please provide ethical approval information regarding blood samples from research team.
Page 1, line 24 The PRP is supposed injectable. Please elaborate the site of injection (venous?) and it’s relationship between the site of biofilm formation.
Author Response
We really appreciate the valuable comments from Reviewer #3. We have clarified several discussions based on the comments. Please find our responses in red color. We also highlighted the major changes in this revision, so that the Editor and Reviewer could easily find them.
Figure 1 -4 Bar chart should be used instead of joining the points of with relevance. Any statistical significance should be indicated on the bars too. Nature of error bar and number of replicate should be shown as well.
Authors response 1: The graphics’ style has been changed. The missing information has been added to the graphics.
Table 1 Unit of the number not indicated throughout the Table. The number of decimal places is not consistent in the Table. Any statistical analysis has been done?
Authors response 2: We modified according to the suggestions. The statistical analyzes were performed with the program GrapPad Prism 8. The results in the Table are expressed as means with standard deviation.
Page 11. 4.2 Please provide ethical approval information regarding blood samples from research team.
Authors response 3: The approval from the ethics committee was given in the article under materials and methods.
"The experiment was approved by the Institutional Ethical Board of Iuliu Hatieganu University of Medicine and Pharmacy, Cluj-Napoca (No. 281/05.07.2018). The study was performed complying with in the Declaration of Helsinki on experimentation involving human subjects".
Page 1, line 24 The PRP is supposed injectable. Please elaborate the site of injection (venous?) and it’s relationship between the site of biofilm formation.
Our study is an experimental study, and the goal was to identify substances with antimicrobial and antibiofilm potential. Obtaining iPRF is an easy procedure; it does not require the use of anticoagulant for preparation and has a higher growth factor content. Our study did not focus on the possibility of IPRF injection, but on the evaluation of the antimicrobial and antibiofilm potential of this substance, and in a mixture with lactoferrin. Its clinical application is not necessarily injectable, more often than not it is used topically (e.g. in superinfected skin wounds, mucosal lesions, together with implants to stimulate regeneration and prevent bacterial superinfections etc).
We have added this a short explanation in the paper.
On behalf of the authors of the manuscript, I thank you for the consideration to our work and the Reviewer 3 for the suggestions that helped to significantly improve the quality of the manuscript.
Sincerely yours,
Emoke Pall
Round 2
Reviewer 2 Report
Thanks for the corrections.